# KINEMATICS-INFORMED REINFORCEMENT LEARNING FOR TRAJECTORY OPTIMIZATION IN CNC MACHINING

## ABSTRACT

Toolpath smoothing and feedrate planning are key techniques in Computer Numerical Control (CNC) machining, and play a significant role in machining accuracy, efficiency, and tool life. Traditional methods typically decouple path smoothing from feedrate planning, without considering the kinematic constraints during the smoothing process. As a result, the subsequent feedrate planning process is subject to more stringent kinematic limitations, which hinders the achievement of optimal speed execution. However, the integration of these two processes presents a significant challenge due to severe complexity and nonlinearity of the problem. Here, we propose a novel Reinforcement Learning (RL) based method, termed KIRL, to address the integrated optimization problem. Experimental results demonstrate that KIRL can generate smoother trajectories and optimize machining time compared to traditional decoupled methods. To our best knowledge, KIRL is the first RL-based method for solving the integrated toolpath smoothing and feedrate planning optimization problem in CNC machining.

## 1 INTRODUCTION

*Computer Numerical Control* (CNC) machining is a widely used manufacturing technique for producing high-precision parts and components across various industries, including aerospace (Nabhani, 2001), automotive (Kim & Song, 2013), and medical devices (Lepasepp & Hurst, 2021). *Toolpath smoothing* and *feedrate planning* are two critical factors that significantly impact machining accuracy, efficiency, and tool life Zhang & Xu (2021). Toolpath smoothing aims to generate a smooth and continuous trajectory for the cutting tool, while feedrate planning determines the speed along the toolpath to minimize machining time while satisfying various kinematic constraints such as maximum velocity, acceleration, and jerk (Altintas & Erkorkmaz, 2003; Beudaert et al., 2012).

In practice, toolpaths are commonly represented using G01 codes (Tulsyan & Altintas, 2015), which consist of a series of continuous line segments. However, linear toolpaths have discontinuities in tangent and curvature at the junctions, which typically result in low machining efficiency and machine vibration (Zhao et al., 2013). To mitigate this issue, path smoothing methods are often employed to create smooth transitions at each corner of the polyline path, followed by velocity planning along the smoothed path curve.

However, the *decoupled* approach often yields suboptimal results, as the smoothed path may not consider the machine tool's *kinematic constraints*, limiting the achievable feedrate in the subsequent planning stage. Recent studies have formulated the integration of toolpath smoothing and feedrate planning into a holistic optimization problem to address this limitation (Yang et al., 2015; Lin et al., 2019; Wu et al., 2023). By considering kinematic constraints during the smoothing process, the integrated approach generates toolpaths that are more suitable for high-speed execution. Nevertheless, solving this integrated optimization problem is quite challenging due to its high nonconvexity and nonlinearity.

In this paper, we propose a novel approach called KIRL (Kinematics-Informed Reinforcement Learning), which leverages Reinforcement Learning (RL) to solve the integrated toolpath smoothing and feedrate planning problem in CNC machining. RL has emerged as a powerful machine learning technique for solving complex *decision-making* problems in diverse domains such as robotics, game

playing, and autonomous driving (Silver et al., 2016; Mnih et al., 2015; Lillicrap et al., 2019). Here, we formulate the integrated optimization problem as a Markov Decision Process (MDP), where each state encapsulates the current kinematic and positional information of the tool, actions correspond to adjustments in kinematic states for the next segment, and rewards are designed to balance machining time and trajectory smoothness. We employ Proximal Policy Optimization (PPO) and Soft Actor-Critic (SAC) (Haarnoja et al., 2018) to train RL agents that predict intermediate kinematic states. Our experimental demonstrations show that KIRL can generate smoother trajectories and optimize machining time effectively compared to traditional decoupled methods.

To summarize, the key contributions of this work are as follows:

- We propose KIRL, the first RL-based method for solving the integrated toolpath smoothing and feedrate planning problem in CNC machining.
- We formulate the integrated optimization problem as an MDP and use PPO and SAC to train RL agents to predict intermediate kinematic states.
- We demonstrate the effectiveness of KIRL in a series of simulated CNC machining tasks, showing improved performance in trajectory smoothness and machining efficiency compared to traditional methods.

## 2 PRELIMINARIES

### 2.1 TOOLPATH REPRESENTATION

**Linear Toolpath.** In CNC machining, toolpaths are often defined as a sequence of discrete points based on G01 commands, which describe straight-line movements. As illustrated in Fig. 1, let $\{\mathbf{p}_i\}_{i=0}^N$ denote a sequence of $N+1$ code points in $\mathbb{R}^2$. The complete linear toolpath $\mathcal{P}$ consists of $N$ linear segments, expressed as:

$$\mathcal{P} = \bigcup_{i=1}^{N} \mathcal{P}_i,$$

where each segment $\mathcal{P}_i, i = 1, \cdots, N$, connects points $\mathbf{p}_{i-1}$ and $\mathbf{p}_i$, parameterized by the scalar $\tau \in [0, 1]$:

$$\mathcal{P}_i = \left\{ \mathbf{p} \in \mathbb{R}^2 : \mathbf{p} = (1-\tau)\mathbf{p}_{i-1} + \tau\mathbf{p}_i \right\}.$$

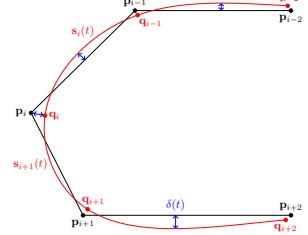

Figure 1: Comparison of linear and smoothed toolpaths.

While this piecewise linear representation ensures geometric continuity, it introduces *discontinuities* in the velocity vector at the *junctions* $\mathbf{p}_i$. These abruptly cause significant changes in velocity direction, particularly in high-speed machining. Sharp turns cause excessive tool wear, vibrations, and necessitate deceleration at each junction, which negatively affects machining efficiency and surface quality.

**Smoothed Toolpath.** To address the issues of linear toolpath, a smoothed toolpath $\mathbf{s}(t)$ is constructed to approximate the original path $\mathcal{P}$ while ensuring smooth transitions between segments. As presented in Fig. 1, $\mathbf{s}(t)$ is defined as a piecewise function over the time interval $[0, T_m]$, where $T_m$ represents the total machining time. Each segment of the smoothed toolpath, $\mathbf{s}_i(t)$, is defined over a time interval $[t_{i-1}, t_i]$ between points $\mathbf{q}_{i-1}$ and $\mathbf{q}_i$:

$$\mathbf{s}(t) = \mathbf{s}_i(t - t_{i-1}), \quad t \in [t_{i-1}, t_i], \quad i \in \{1, \ldots, N\}.$$

The points $\mathbf{q}_i$, typically located near the original G01 points $\mathbf{p}_i$, are chosen to minimize the discrepancy between the original and smoothed paths while ensuring continuity in velocity and acceleration. This smoothing reduces tool wear and vibrations, and thus allows for higher machining speeds.

**Chord Error.** The accuracy of the smoothed toolpath is evaluated by the chord error, which measures the perpendicular distance between the smoothed path $\mathbf{s}(t)$ and the set of the original linear segments $\mathcal{P}$. For each smoothed segment $\mathbf{s}_i(t)$, the chord error $\delta_i(t)$ is calculated by

$$\delta_i(t) = \frac{\|(\mathbf{s}_i(t) - \mathbf{p}_{i-1}) \times (\mathbf{p}_i - \mathbf{p}_{i-1})\|}{\|\mathbf{p}_i - \mathbf{p}_{i-1}\|}.$$

Consequently, minimising the chord error ensures that the smoothed path $\mathbf{s}(t)$ remains close to the original toolpath $\mathcal{P}$, while maintaining a balance between machining accuracy and efficiency.

## 2.2 CONSTRAINTS

The optimization process must adhere to several constraints to ensure the generated trajectory is both feasible and secure for CNC operations.

**Kinematic Constraints.** The trajectory must comply with the machine's physical limits on velocity, acceleration, and jerk. These constraints are defined as:

$$\|\dot{\mathbf{s}}_i(t)\| \leq v_{\max}, \quad \|\ddot{\mathbf{s}}_i(t)\| \leq a_{\max}, \quad \|\dddot{\mathbf{s}}_i(t)\| \leq j_{\max}, \quad i \in \{1, \ldots, N\}, \tag{1}$$

where $v_{\max}$, $a_{\max}$, and $j_{\max}$ represent the maximum allowable velocity, acceleration, and jerk, respectively.

**Chord Error Constraint.** In addition to the kinematic constraints, the smoothed toolpath must adhere to a chord error constraint, ensuring the toolpath's deviation from the original linear path does not exceed a predefined tolerance:

$$\delta_i(t) \leq \delta_{\max}, \quad i \in \{1, \ldots, N\}, \tag{2}$$

where $\delta_{\max}$ is the maximum allowable chord error. This constraint ensures geometric accuracy and limits toolpath deviation during machining.

## 2.3 OPTIMIZATION OBJECTIVES

The trajectory optimization problem aims to balance the goals of improving machining efficiency while ensuring trajectory smoothness.

**Trajectory Jerk Minimization.** To achieve a smooth trajectory, we need to minimize the integral of the squared jerk $J$. Jerk, denoted by $\dddot{\mathbf{s}}(t)$, represents the third derivative of position $\mathbf{s}(t)$ with respect to time $t$. The jerk minimization objective is formulated as:

$$J = \sum_{i=1}^{N} J_i = \sum_{i=1}^{N} \int_{t_{i-1}}^{t_i} \|\dddot{\mathbf{s}}(t)\|^2 \, \mathrm{d}t = \sum_{i=1}^{N} \int_{0}^{T_i} \|\dddot{\mathbf{s}}_i(t)\|^2 \, \mathrm{d}t, \tag{3}$$

where $T_i = t_i - t_{i-1}$ is the duration of the $i$-th segment. Minimizing the jerk ensures that the toolpath is smooth and continuous, reducing udden changes in motion that can lead to tool wear, vibrations, and potential inaccuracies in the machined part.

**Machining Time Minimization.** The second key objective is minimizing the total machining time $T_m$, which is the cumulative time for all segments:

$$T_m = \sum_{i=1}^{N} T_i. \tag{4}$$

Reducing machining time enhances production efficiency by lowering overall cycle times while maintaining quality and precision.

## 3 PROPOSED METHOD

According to the above preliminaries, our target is to perform an integrated optimization to find a trajectory $\mathbf{s}(t)$ that minimizes a weighted sum of both the trajectory jerk and the machining time. This optimization process takes into account the kinematic constraints:

$$\min_{\mathbf{s}(t)} \quad J + wT_m = \int_{0}^{T_m} \|\dddot{\mathbf{s}}(t)\|^2 + w \, \mathrm{d}t, \tag{5}$$

$$\text{s.t.} \quad \text{Constraints in Eqs. (1) and (2) are satisfied.}$$

Here $w$ is the weighting factor that balances the importance of each objective in the optimization process.

### 3.1 TOOLPATH SEGMENTATION

To solve Eq. (5), we first divide the toolpath into $N$ segments. Each segment corresponds to the portion of the toolpath between two successive toolpath points. As illustrated in Fig. 2, the $i$-th segment, denoted as $\mathbf{s}_i(t)$, connects the boundary points $\mathbf{q}_{i-1}$ and $\mathbf{q}_i$.

For each segment, we consider the duration $T_i$ and specify the kinematic states at the segment boundaries, which include positions $\mathbf{q}_{i-1}$ and $\mathbf{q}_i$, velocities $\mathbf{v}_{i-1}$ and $\mathbf{v}_i$, and accelerations $\mathbf{a}_{i-1}$ and $\mathbf{a}_i$. These boundary conditions ensure smooth transitions between segments in terms of position, velocity, and acceleration.

The kinematic state constraints at the segment boundaries are expressed as:

$$\begin{aligned}
\mathbf{s}_i(0) &= \mathbf{q}_{i-1}, \quad \mathbf{s}_i(T_i) = \mathbf{q}_i, \\
\dot{\mathbf{s}}_i(0) &= \mathbf{v}_{i-1}, \quad \dot{\mathbf{s}}_i(T_i) = \mathbf{v}_i, \\
\ddot{\mathbf{s}}_i(0) &= \mathbf{a}_{i-1}, \quad \ddot{\mathbf{s}}_i(T_i) = \mathbf{a}_i.
\end{aligned} \tag{6}$$

The optimization problem for the $i$-th segment is then formulated as:

$$\min_{\mathbf{s}_i(t)} \quad \int_0^{T_i} \| \dddot{\mathbf{s}}_i(t) \|^2 \, \mathrm{d}t, \tag{7}$$

$$\text{s.t.} \quad \text{Constraints in Eqs. (1), (2) and (6) are satisfied.}$$

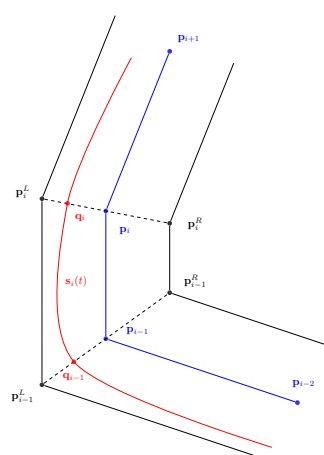

Figure 2: Toolpath segmentation approach. The blue line represents the linear toolpath, and the red curve represents the smoothed toolpath. The line segment $\overline{\mathbf{p}_i^L \mathbf{p}_i^R}$ represents the angle bisector of $\angle \mathbf{p}_{i-1} \mathbf{p}_i \mathbf{p}_{i+1}$.

The objective is to minimize the integral of the squared jerk over the duration $T_i$, which promotes smoothness in the trajectory. The constraints ensure that the smoothed path adheres to the maximum allowable chord error $\delta_{\max}$, complies with the machine's kinematic limits, and satisfies the boundary conditions for continuity.

After theoretical analysis, we establish the following theorem:

**Theorem 1.** *The optimal solution to the optimization problem in Equation equation 7 is a quintic polynomial function of time $t$.*

*Proof.* The proof of this theorem can be found in Appendix A. $\qquad\square$

By employing quintic polynomials for each segment, we ensure that the trajectory is continuous and differentiable up to the second derivative, satisfying the position, velocity, and acceleration constraints at the boundaries. This approach yields a smooth and feasible toolpath that minimizes jerk while respecting all physical and geometrical limitations.

### 3.2 MOTION PRIMITIVE GENERATION

From Theorem 1, we define the quintic polynomial function of time $t$ explicitly as:

$$\mathbf{s}_i(t) = \mathbf{c}_{i0} + \mathbf{c}_{i1}t + \mathbf{c}_{i2}t^2 + \mathbf{c}_{i3}t^3 + \mathbf{c}_{i4}t^4 + \mathbf{c}_{i5}t^5, \quad t \in [0, T_i], \tag{8}$$

where $\mathbf{c}_{i0}, \mathbf{c}_{i1}, \ldots, \mathbf{c}_{i5}$ are the coefficients of the quintic polynomial. The coefficients can be determined by imposing the boundary conditions Eq. (6). We can transform the boundary conditions into a linear system of equations:

$$\mathbf{A}\mathbf{c} = \mathbf{q}, \tag{9}$$

where $\mathbf{A} \in \mathbb{R}^{6 \times 6}$ represents the coefficient matrix that encodes the time information (detailed in Appendix B), $\mathbf{c} = [\mathbf{c}_{i0}\mathbf{c}_{i1}, \mathbf{c}_{i2}, \mathbf{c}_{i3}, \mathbf{c}_{i4}, \mathbf{c}_{i5}]^\top$, and $\mathbf{q} = [\mathbf{q}_{i-1}, \mathbf{v}_{i-1}, \mathbf{a}_{i-1}, \mathbf{q}_i, \mathbf{v}_i, \mathbf{a}_i]^\top$.

To accelerate the computation of the quintic polynomial coefficients, we precompute $\mathbf{A}^{-1}$, allowing us to efficiently solve for the coefficients using matrix multiplication. From the precomputed $\mathbf{A}^{-1}$,

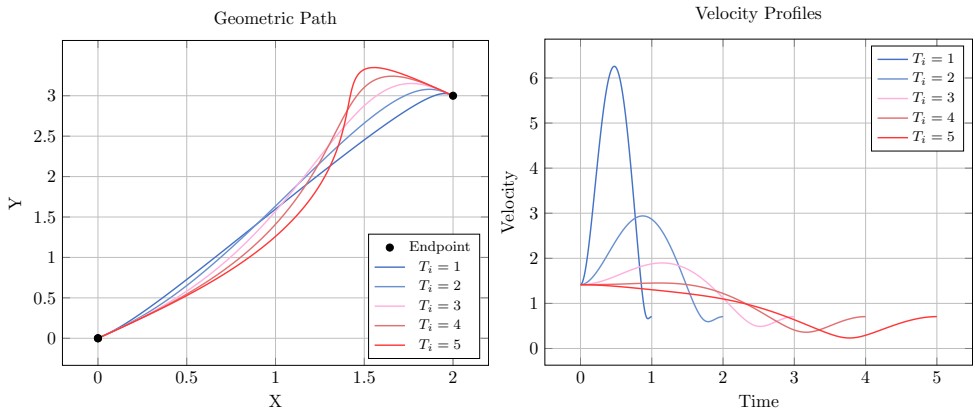

Figure 3: Left: 2D trajectories with different $T_i$ values. Right: Velocity profiles of the trajectories.

we conclude that the coefficients are determined exclusively by the segment duration $T_i$ when the boundary conditions are fixed.

As illustrated in Fig. 3, increasing the segment duration $T_i$ results in more distorted trajectories, while the velocity profile exhibits greater smoothness. Therefore, the segment duration $T_i$ serves as a hyperparameter to balance the trade-off between trajectory smoothness and velocity profile smoothness.

### 3.3 REINFORCEMENT LEARNING FOR KINEMATIC STATE PREDICTION

In the previous section, we have presented a method for generating smooth trajectories within each segment using quintic polynomial functions. However, the kinematic states at the segment boundaries are still unknown. To predict the optimal kinematic state at the boundary of each segment, we use an RL agent trained by the Proximal Policy Optimization (PPO) algorithm (Schulman et al., 2017) and Soft Actor-Critic (SAC) algorithm (Haarnoja et al., 2018). Within the current segment, assuming the initial kinematic state is $(\mathbf{q}_{i-1}, \mathbf{v}_{i-1}, \mathbf{a}_{i-1})$, then the RL agent is trained to predict the optimal kinematic state $(\mathbf{q}_i^{\text{pred}}, \mathbf{v}_i^{\text{pred}}, \mathbf{a}_i^{\text{pred}})$ at the end of the segment.

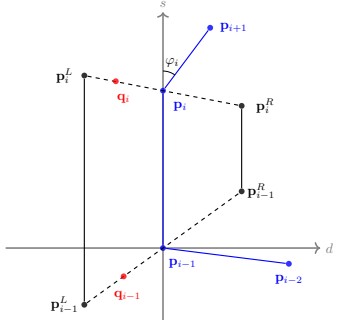

Figure 4: Local coordinate system of the segment.

To enhance the generalization ability of the target RL agent, we use the local coordinate system of the segment to represent the kinematic states. As shown in Fig. 4, the local coordinate system originates from $\mathbf{p}_{i-1}$, with the $s$-axis extending from $\mathbf{p}_{i-1}$ to $\mathbf{p}_i$, and the $d$-axis being orthogonal to the $s$-axis. The transformation from the global to the local coordinate system is mathematically expressed as

$$\tilde{\mathbf{p}} = \mathbf{R}(\theta_i - \frac{\pi}{2})^\top (\mathbf{p} - \mathbf{p}_i), \quad \mathbf{R}(\theta) = \begin{bmatrix} \cos\theta & -\sin\theta \\ \sin\theta & \cos\theta \end{bmatrix}, \tag{10}$$

where $\mathbf{p}$ and $\tilde{\mathbf{p}}$ are the coordinates in the global and local coordinate systems, respectively, and $\theta_i = \arctan(\mathbf{p}_i - \mathbf{p}_{i-1})$ represents the angle between the $s$-axis and the $x$-axis.

**Observation Space.** The observation space describes the current state of the system that the RL agent can observe. In our RL environment, the state vector $\mathsf{S}_i \in \mathbb{R}^{10}$ is expressed as

$$\mathsf{S}_i = [N - i, \ell_i, \ell_{i+1}, \varphi_i, \tilde{\mathbf{q}}_{i-1}^\top, \tilde{\mathbf{v}}_{i-1}^\top, \tilde{\mathbf{a}}_{i-1}^\top]^\top, \tag{11}$$

where $N$ denotes the total number of segments, $\ell_i = \|\mathbf{p}_i - \mathbf{p}_{i-1}\|$ represents the segment length, $\varphi_i$ is the turning angle between $\overrightarrow{\mathbf{p}_{i-1}\mathbf{p}_i}$ and $\overrightarrow{\mathbf{p}_i\mathbf{p}_{i+1}}$, and $\tilde{\mathbf{q}}_{i-1}, \tilde{\mathbf{v}}_{i-1}$, and $\tilde{\mathbf{a}}_{i-1}$ denote the position, velocity, and acceleration in the local coordinate system, respectively.

**Action Space.** For each observation $\mathsf{S}_i$, the RL agent produces an action $\mathsf{A}_i \in \mathbb{R}^5$, which predicts the kinematic state at the end of the segment. To ensure that $\mathbf{q}_i^{\text{pred}}$ lies on the line segment $\overline{\mathbf{p}_i^L \mathbf{p}_i^R}$, we parameterize $\tilde{\mathbf{q}}_i^{\text{pred}}$ by $u_i$ as follows:

$$\tilde{\mathbf{q}}_i^{\text{pred}} = \tilde{\mathbf{p}}_i + u_i \left( \tilde{\mathbf{p}}_i^R - \tilde{\mathbf{p}}_i \right), \tag{12}$$

where $\tilde{\mathbf{p}}_i^R$ denotes the right endpoint of the segment. Then the action vector is defined as:

$$\mathsf{A}_i = [u_i, \tilde{\mathbf{v}}_i^{\text{pred}}, \tilde{\mathbf{a}}_i^{\text{pred}}], \tag{13}$$

After receiving the action, the trajectory $\mathbf{s}_i(t)$ can be generated using the quintic polynomial function with $T_i$.

**Reward Function.** When action $\mathsf{A}_i$ is executed, the RL agent receives a reward signal $\mathsf{R}_i$, which evaluates the quality of the generated trajectory. The reward function is defined as

$$\mathsf{R}_i = \lambda_1 r_i^{\text{time}} + \lambda_2 r_i^{\text{jerk}} + \lambda_3 r_i^{\text{chord}} + r_i^{\text{constraint}}, \tag{14}$$

in which $r_i^{\text{time}}$ rewards the reduction in machining time, $r_i^{\text{constraint}}$ penalizes violations of the kinematic constraints, and $r_i^{\text{chord}}$ penalizes deviations from the desired path. Concretely, the components of the reward are defined as follows:

$$r_i^{\text{time}} = T_i^{\text{pred}}, \quad r_i^{\text{jerk}} = \int_0^{T_i} \left[ j_i(t) - j_{\max} \right]_+ \, \mathrm{d}t, \quad r_i^{\text{chord}} = \int_0^{T_i} \left( \frac{\delta_i(t)}{\delta_{\max}} \right)^2 \, \mathrm{d}t,$$

$$r_i^{\text{constraint}} = \begin{cases} -C_{\text{penalty}} & \text{if any } v_i(t), a_i(t), j_i(t), \text{ or } \delta_i(t) \text{ exceeds its limit,} \\ 0 & \text{otherwise,} \end{cases}$$

where $v_i(t) = \|\dot{\mathbf{s}}_i(t)\|$ is the velocity, $a_i(t) = \|\ddot{\mathbf{s}}_i(t)\|$ is the acceleration, and $j_i(t) = \|\dddot{\mathbf{s}}_i(t)\|$ is the jerk. If any of the kinematic constraints are violated, the reinforcement learning process is terminated immediately, and a large penalty $C_{\text{penalty}}$ is applied. The weights $\lambda_1$, $\lambda_2$, and $\lambda_3$ are hyperparameters that balance the importance of each component in the reward function.

**Optimal Duration Search.** Before evaluating the generated trajectory using the reward function, we need to determine the appropriate duration $T_i$. When the initial and final kinematic states are fixed, the reward function $\mathsf{R}_i$ becomes a univariate function with respect to $T_i$. Hence, we aim to minimize the negative reward function, which can be formulated as:

$$T_i = \arg\min_T -\mathsf{R}_i(T). \tag{15}$$

This optimization problem is a *bounded one-dimensional minimization problem*, which can be efficiently solved using Brent's method (Brent, 1971). Brent's method combines the robustness of the golden-section search with the speed of parabolic interpolation. It dynamically switches between these strategies based on the function's behavior to ensure both stability and rapid convergence. Importantly, it does not rely on derivative information, making it particularly suitable for optimizing non-smooth or complex reward functions in our framework.

**Overall Algorithm** The trajectory generation process iterates through each segment, utilizing the RL agent to predict optimal kinematic states, optimizing the segment duration, generating the trajectory using quintic polynomials, and ensuring all constraints are satisfied. The algorithm is presented below:

---

**Algorithm 1:** Reinforcement Learning-Based Trajectory Generation

---

**Input** : Waypoints: $\mathbf{p}_0, \mathbf{p}_1, \ldots, \mathbf{p}_N$
Pre-trained RL Agent (PPO or SAC)
Hyperparameters: $\lambda_1, \lambda_2, \lambda_3, C_{\text{penalty}}$
Constraints: $v_{\max}, a_{\max}, j_{\max}, \delta_{\max}$

**Output:** Generated Trajectory: $\mathbf{s}(t)$

1 Initialize $\mathbf{q}_0 = \mathbf{p}_0, \mathbf{v}_0 = \mathbf{0}, \mathbf{a}_0 = \mathbf{0}$;
2 **for** $i \leftarrow 1$ **to** $N$ **do**
3     **if** $i > 1$ **then**
4         // Prepare observation
5         $\mathsf{S}_i = \text{ConstructObservation}(i, \mathbf{q}_{i-1}, \mathbf{v}_{i-1}, \mathbf{a}_{i-1})$;
6         // RL Prediction
7         $\mathsf{A}_i = \text{RL Agent}(\mathsf{S}_i)$;
8         // Set boundary conditions
9         $\mathbf{q}_i, \mathbf{v}_i, \mathbf{a}_i = \text{SetBoundary}(\mathsf{A}_i)$;
10     **end if**
11     // Optimize Duration
12     $T_i = \text{OptimizeDuration}(\mathbf{q}_{i-1}, \mathbf{q}_i)$;
13     // Generate Trajectory Segment
14     $\mathbf{s}_i(t) = \text{QuinticPolynomial}(\mathbf{q}_{i-1}, \mathbf{v}_{i-1}, \mathbf{a}_{i-1}, \mathbf{q}_i, \mathbf{v}_i, \mathbf{a}_i, T_i)$;
15     // Evaluate Constraints
16     **if** *Constraints Violated* **then**
17         **Terminate** and Apply Penalty;
18     **end if**
19     // Append to Trajectory
20     $\mathbf{s}(t) \leftarrow \mathbf{s}(t) \cup \mathbf{s}_i(t)$;
21 **end for**
22 // Finalize Trajectory
23 **if** *Global Constraints Satisfied* **then**
24     **return** $\mathbf{s}(t)$;
25 **end if**
26 **else**
27     **return** Failure;
28 **end if**

---

## 4 EXPERIMENTS

In this section, we evaluate the performance of the proposed KIRL method on common CNC machining tasks. As the first RL-based method in this domain, we compare KIRL with representative traditional methods and analyze the results based on multiple metrics related to *machining time*, *trajectory smoothness*, and *kinematic performance*.

### 4.1 EXPERIMENTAL SETUP

**Datasets and Preprocessing.** We adopt four representative toolpaths for test: *Butterfly*, *Dolphin*, *Golden Fish*, and *Shark*. These toolpaths are obtained from a publicly available repository[1] and were selected for their varying complexity and curvature characteristics. This provides a comprehensive evaluation of the methods under different conditions. The original linear toolpaths are normalized to fit within a $[0, 100]^2$ coordinate space to ensure consistency across different paths. Each toolpath is resampled to generate 200 equally spaced points, serving as the input format for all approaches.

**Compared Methods.** We compare KIRL against two traditional decoupled approaches for toolpath smoothing and feedrate planning: 1) **ICR method** (Zhao et al., 2013): An inscribed corner rounding (ICR) technique that smooths toolpaths using curvature-continuous B-splines with G2 continuity and performs feedrate planning; 2) **CCR method** (Xu & Sun, 2018): A circumscribed

---

[1]The raw NC G-code files are obtained from https://cncgcode.weebly.com/

corner rounding (CCR) technique that smooths the toolpath by inserting transition curves using double cubic B-splines at junctions, followed by conventional feedrate planning. Both approaches utilize a jerk-limited S-shape acceleration-deceleration algorithm for feedrate planning (Lin et al., 2007). For our proposed KIRL method, we implement two variants using different reinforcement learning algorithms: 1) **KIRL-PPO**: KIRL using Proximal Policy Optimization (PPO) (Schulman et al., 2017); 2) **KIRL-SAC**: KIRL using Soft Actor-Critic (SAC) (Haarnoja et al., 2018). Due to the unavailability of baseline implementations, we have re-implemented these approaches ourselves. We plan to release all of the code to facilitate future research in learning-guided CNC maching.

## 4.2 EVALUATION METRICS

To comprehensively assess the performance of all compared methods, we employ the following metrics for evaluation. 1) **Total Machining Time** ($T_m$): The total time required to complete the whole toolpath, reflecting machining efficiency. 2) **Maximum Curvature** ($\kappa_{\max}$): The highest curvature along the toolpath, indicating the sharpest turn impacting machining quality; **Maximum Turning Angle** ($\theta_{\max}$): The largest angle between consecutive path segments, indicating trajectory smoothness. These two metrics are used for path smoothness assessment. 3) For kinematic smoothness evaluation, we use both **RMS Acceleration** ($a_{\mathrm{rms}}$) and **RMS Jerk** ($j_{\mathrm{rms}}$). $a_{\mathrm{rms}}$ is defined as the variability in acceleration along the toolpath computed as:

$$a_{\mathrm{rms}} = \sqrt{\frac{1}{N} \sum_{i=1}^{N} \|\mathbf{a}_i\|^2}, \tag{16}$$

where $N$ is the number of points and $\mathbf{a}_i$ is the acceleration at point $i$. $j_{\mathrm{rms}}$ is the change rate of acceleration (jerk) computed as:

$$j_{\mathrm{rms}} = \sqrt{\frac{1}{N} \sum_{i=1}^{N} \|\mathbf{j}_i\|^2}, \tag{17}$$

where $\mathbf{j}_i$ is the jerk at point $i$. Lower values of $\kappa_{\max}$, $\theta_{\max}$, $a_{\mathrm{rms}}$, and $j_{\mathrm{rms}}$ indicate smoother trajectories and better kinematic performance, while a lower $T_m$ indicates higher machining efficiency.

## 4.3 EXPERIMENTAL RESULTS AND DISCUSSION

**Numerical Results.** Tab. 1 reports the results for each toolpath, with the best-performing method highlighted in bold. As observed, on the *Butterfly* toolpath, KIRL-PPO achieves the lowest maximum curvature, indicating a smoother path compared to the baselines. KIRL-SAC attains the lowest RMS acceleration and jerk, reflecting better kinematic smoothness. Both KIRL variants outperform the baselines in most metrics, demonstrating the effectiveness of integrating kinematic constraints during optimization.

For the *Dolphin* toolpath, KIRL-SAC achieves the best performance in maximum curvature, turning angle, RMS acceleration, and jerk. Although ICR method achieve a slightly lower machining time, KIRL-SAC provides a better balance between efficiency and smoothness.

On the *Golden Fish* toolpath, KIRL-PPO excels in minimizing the maximum turning angle, RMS acceleration, and jerk. While ICR method achieve the lowest machining time, the trajectories generated by KIRL are smoother, which can enhance machining quality and reduce tool wear.

For the *Shark* toolpath, KIRL-PPO outperforms all baselines across all metrics. Notably, it achieves a significant reduction in maximum curvature and kinematic quantities, leading to smoother and more efficient machining processes.

**Trajectory Visualization.** Fig. 5 illustrates the trajectories generated by KIRL-PPO and ICR methods on the *Shark* toolpath, as well as KIRL-SAC and CCR methods on the *Dolphin* toolpath. Across both toolpaths, the KIRL-based methods produce smoother transitions and fewer abrupt changes in direction compared to the traditional methods, highlighting their effectiveness in generating more refined and continuous trajectories.

Table 1: Performance Comparison Across Different Toolpaths. The best-performing method for each metric under each toolpath is highlighted in **bold**.

| Toolpath | Metric | KIRL-PPO | KIRL-SAC | CCR | ICR |
|---|---|---|---|---|---|
| *Butterfly* | Maximum Curvature | **25.61** | 363.95 | 45.72 | 80.52 |
| | Maximum Turning Angle | 0.0502 | 0.0735 | **0.0295** | 0.0335 |
| | RMS Acceleration | 52.46 | **37.78** | 85.64 | 78.25 |
| | RMS Jerk | 1655.67 | **1025.50** | 5723.21 | 5405.78 |
| | Machining Time | **41.16** | 45.37 | 43.38 | 42.36 |
| *Dolphin* | Maximum Curvature | 17.37 | **16.41** | 36.90 | 27.11 |
| | Maximum Turning Angle | 0.0204 | **0.0137** | 0.0263 | 0.0217 |
| | RMS Acceleration | 59.53 | **56.85** | 108.05 | 89.03 |
| | RMS Jerk | 2377.80 | **2341.08** | 7134.80 | 6326.47 |
| | Machining Time | 31.76 | 31.02 | 31.81 | **29.71** |
| *Golden Fish* | Maximum Curvature | 9.23 | **6.23** | 31.14 | 25.88 |
| | Maximum Turning Angle | **0.0096** | 0.0157 | 0.0258 | 0.0225 |
| | RMS Acceleration | **42.57** | 45.67 | 93.72 | 84.99 |
| | RMS Jerk | **1163.34** | 1367.33 | 6292.79 | 5788.44 |
| | Machining Time | 45.46 | 43.88 | 43.39 | **42.08** |
| *Shark* | Maximum Curvature | **7.30** | 24.83 | 36.52 | 48.91 |
| | Maximum Turning Angle | **0.0204** | 0.0336 | 0.0271 | 0.0272 |
| | RMS Acceleration | **63.04** | 88.84 | 123.25 | 108.65 |
| | RMS Jerk | **2766.11** | 4002.50 | 7883.43 | 7388.95 |
| | Machining Time | **27.87** | 28.44 | 30.96 | 28.94 |

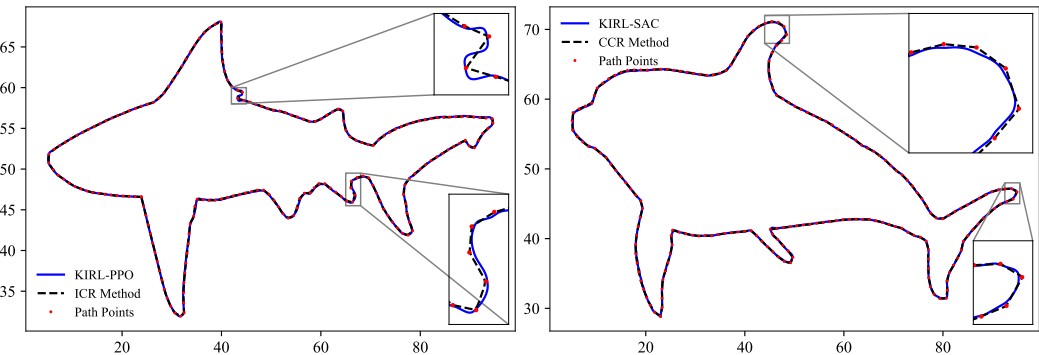

Figure 5: Trajectory comparison of KIRL-PPO against the ICR Method for the *Shark* toolpath (left), and KIRL-SAC against the CCR Method for the *Dolphin* toolpath (right). Red dots indicate the input toolpath points.

**Discussion.** The experimental results consistently show that KIRL generates smoother trajectories compared to traditional methods by integrating kinematic constraints during optimization. The RL-based approach allows KIRL to adaptively predict intermediate kinematic states that adhere to the machine's limitations, resulting in improved kinematic smoothness without compromising efficiency. Although some baselines achieve slightly shorter machining times, this comes at the cost of increased acceleration and jerk, which can lead to machine wear and reduced machining quality. KIRL provides a balanced trade-off between efficiency and smoothness, highlighting the advantages of the integrated optimization approach.

### 4.4 LIMITATIONS AND FUTURE WORK

**Limitations.** KIRL demonstrates promising results, but several limitations need to be addressed for broader industrial adoption. First, the training process is slow due to the CPU-based simulation environment, resulting in long training times and low GPU utilization. Second, KIRL requires sep-

arate training for each toolpath, which limits its scalability, especially in environments with many unique toolpaths. Additionally, the current implementation only supports 2-axis machining, which restricts its applicability to more advanced multi-axis operations. Lastly, KIRL has only been evaluated in simulation, with no real-world validation on physical CNC machines.

**Future Work.** Future efforts will focus on several key areas to overcome these limitations. First, improving training efficiency will be critical—this could be achieved by incorporating parallel environment sampling to make better use of available computational resources and reduce training times. Second, enabling KIRL to generalize across different toolpaths through transfer learning or meta-learning could significantly improve scalability and reduce the need for retraining. Expanding KIRL to support multi-axis machining, such as 5-axis and 6-axis operations, will enhance its industrial relevance by addressing more complex machining tasks. Finally, real-world validation on actual CNC machines will be essential to assess KIRL's performance, reliability, and potential challenges in practical settings.

## 5 CONCLUSIONS

In this work, we introduced KIRL, a Reinforcement Learning-based approach for the integrated optimization of toolpath smoothing and feedrate planning in CNC machining. By formulating the problem as a Markov Decision Process and utilizing advanced RL algorithms like PPO and SAC, KIRL effectively predicts intermediate kinematic states that balance machining efficiency with trajectory smoothness. Experimental results demonstrated that KIRL outperforms traditional decoupled methods in generating smoother trajectories and optimizing machining time across various complex toolpaths.

## REPRODUCIBILITY STATEMENT

To the best of our knowledge, this work presents the first RL-based method for optimizing the toolpath smoothing and feedrate planning problem in CNC machining. To reproduce the experimental results, we elaborate on the hyper-parameter settings in Appendix C. To facilitate reproducibility, we also publicly release our code.

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

# A    PROOF OF THEOREM 1

*Proof.* Let $\mathbf{s}_i(t)$ be the optimal solution of Eq. (7). Consider an additive perturbation $\mathbf{p}(t)$ that satisfies the boundary conditions $\mathbf{p}(0) = \mathbf{p}(T_i) = \mathbf{0}$, $\dot{\mathbf{p}}(0) = \dot{\mathbf{p}}(T_i) = \mathbf{0}$, and $\ddot{\mathbf{p}}(0) = \ddot{\mathbf{p}}(T_i) = \mathbf{0}$. Then, we have

$$\int_0^{T_i} \| \dddot{\mathbf{s}}_i(t) \|^2 \, \mathrm{d}t \le \int_0^{T_i} \| \dddot{\mathbf{s}}_i(t) + \lambda \dddot{\mathbf{p}}(t) \|^2 \, \mathrm{d}t, \tag{18}$$

and the function

$$\begin{aligned}
f(\lambda) &= \int_0^{T_i} \| \dddot{\mathbf{s}}_i(t) + \lambda \dddot{\mathbf{p}}(t) \|^2 \, \mathrm{d}t \\
&= \int_0^{T_i} \| \dddot{\mathbf{s}}_i(t) \|^2 \, \mathrm{d}t + 2\lambda \int_0^{T_i} \dddot{\mathbf{s}}_i(t) \cdot \dddot{\mathbf{p}}(t) \, \mathrm{d}t \\
&\quad + \lambda^2 \int_0^{T_i} \| \dddot{\mathbf{p}}(t) \|^2 \, \mathrm{d}t
\end{aligned} \tag{19}$$

is minimized at $\lambda = 0$, which implies that

$$f'(0) = 2 \int_0^{T_i} \dddot{\mathbf{s}}_i(t) \cdot \dddot{\mathbf{p}}(t) \, \mathrm{d}t = 0. \tag{20}$$

Using integration by parts, we have

$$\int_0^{T_i} \dddot{\mathbf{s}}_i(t) \cdot \dddot{\mathbf{p}}(t) \, \mathrm{d}t = - \int_0^{T_i} \mathbf{s}_i^{(6)}(t) \cdot \mathbf{p}(t) \, \mathrm{d}t = 0, \tag{21}$$

which implies that $\mathbf{s}_i(t)$ is a polynomial of degree at most five. $\qquad\square$

# B    DETAILS OF THE LINEAR SYSTEM

The boundary conditions defined in Eq. (6) can be transformed into the following linear system:

$$\begin{bmatrix} 1 & 0 & 0 & 0 & 0 & 0 \\ 0 & 1 & 0 & 0 & 0 & 0 \\ 0 & 0 & 2 & 0 & 0 & 0 \\ 1 & T_i & T_i^2 & T_i^3 & T_i^4 & T_i^5 \\ 0 & 1 & 2T_i & 3T_i^2 & 4T_i^3 & 5T_i^4 \\ 0 & 0 & 2 & 6T_i & 12T_i^2 & 20T_i^3 \end{bmatrix} \begin{bmatrix} \mathbf{c}_{i0} \\ \mathbf{c}_{i1} \\ \mathbf{c}_{i2} \\ \mathbf{c}_{i3} \\ \mathbf{c}_{i4} \\ \mathbf{c}_{i5} \end{bmatrix} = \begin{bmatrix} \mathbf{q}_{i-1} \\ \mathbf{v}_{i-1} \\ \mathbf{a}_{i-1} \\ \mathbf{q}_i \\ \mathbf{v}_i \\ \mathbf{a}_i \end{bmatrix}. \tag{22}$$

By precomputing the inverse of the matrix in Eq. (22), we can solve for the coefficients efficiently:

$$\begin{bmatrix} \mathbf{c}_{i0} \\ \mathbf{c}_{i1} \\ \mathbf{c}_{i2} \\ \mathbf{c}_{i3} \\ \mathbf{c}_{i4} \\ \mathbf{c}_{i5} \end{bmatrix} = \begin{bmatrix} 1 & 0 & 0 & 0 & 0 & 0 \\ 0 & 1 & 0 & 0 & 0 & 0 \\ 0 & 0 & 1/2 & 0 & 0 & 0 \\ -10/T_i^3 & -6/T_i^2 & -3/(2T_i) & 10/T_i^3 & -4/T_i^2 & 1/(2T_i) \\ 15/T_i^4 & 8/T_i^3 & 3/(2T_i^2) & -15/T_i^4 & 7/T_i^3 & -1/T_i^2 \\ -6/T_i^5 & -3/T_i^4 & -1/(2T_i^3) & 6/T_i^5 & -3/T_i^4 & 1/(2T_i^3) \end{bmatrix} \begin{bmatrix} \mathbf{q}_{i-1} \\ \mathbf{v}_{i-1} \\ \mathbf{a}_{i-1} \\ \mathbf{q}_i \\ \mathbf{v}_i \\ \mathbf{a}_i \end{bmatrix}. \tag{23}$$

From Eq. (23), we observe that the coefficients depend solely on the segment duration $T_i$ when the boundary conditions are fixed. This insight allows for efficient computation and highlights the role of $T_i$ as a hyperparameter in balancing trajectory smoothness and velocity profile smoothness.

# C    ADDITIONAL EXPERIMENTAL DETAILS

**Experimental Setup.**    For the experimental setup, the following parameters were used:

- Maximum speed: 10
- Maximum acceleration: 100
- Maximum jerk: 10,000
- Chord error tolerance: 0.5
- Interpolation period: 0.0005s

**Training Parameters.**   For training the KIRL agents using PPO and SAC, we set the following hyperparameters:

- Learning rate: $3 \times 10^{-4}$
- Discount factor ($\gamma$): 1
- Number of training epochs: 2000,000
- Batch size: 64
- Network architecture: Three hidden layers with 256 neurons each and ReLU activation.

