# OpenReview forum: "Kinematics-Informed Reinforcement Learning for Trajectory Optimization in CNC Machining"
_ICLR.cc/2025/Conference — ICLR 2025 Conference Withdrawn Submission_

### Official Review · Reviewer_5bTc · 2024-10-27

**Soundness:** 2
**Presentation:** 3
**Contribution:** 1
**Rating:** 1
**Confidence:** 5

**Summary:**

The paper proposes to apply reinforcement learning to jointly solve the feedrate planning and path smoothing problem for CNC machines.
The paper proposes a formulation for the problem, including reward function, state space, and action space/policy parametrization. The proposed approach is compared in simulation on four 2D trajectories against non-RL baselines, and shown to outperform those.

**Strengths:**

The paper is generally well written and does a good job at explaining the problem. The paper tackles and important, practical problem in CNC. The novel method is presented clearly. The method seems correct and sound. The experiments show good results compared to the baselines.

**Weaknesses:**

While the paper tackles and important practical problem, I did not see any methodological novelty. It is an application of very standard RL algorithms to a novel problem. The specific formulation of the optimization objective/reward function, state space, and action space/policy representation seem novel, but I could not extract any more general lessons learned from them. There is nothing per se wrong with the design choices made and the resulting overall system seems to work. However, there are quite a few choices where alternatives would be possible and insights into why certain choices were made are lacking - I would have expected at least some ablations in the experiments. As the paper itself points out, in its current form the results are very far from practical applicability and rather a proof of concept.
I also have a few doubts about details of the method and paper, and about the evaluation. See questions below.

To sum up, overall there is neither an algorithmic contributions, nor sufficient general insights for a systems paper. The experimental results also do not comply with basic standards for RL papers.

**Questions:**

- A motivation is missing why this is treated as a sequential decision making problem rather than a global optimization problem. I'd assume better global performance can be achieved when optimizing over the whole trajectory rather than considering the past as fixed and only considering the next straight segment.
- So more generally: Why is the optimization done per segment? In the initial parts with the optimization problem that can lead to arbitrarily bad performance on subsequent segments, in the RL formulation this is accounted for as it optimizes for the cumulative reward.
- Throughout the paper costs/constraints/optimization objectives are mixed in terms of "global" (whole trajectory) and "local" per segment, e.g. the ones in Eq (3) vs (4). That makes it rather hard to follow.
- The constraints and objectives change throughout the paper - which makes it rather hard to follow. In Sect. 2.2 we have constraints on velocity, acceleration and jerk, in Sect. 3.1 that becomes a minimization of the jerk (while complying to the constraint on chord error, which intuitively would mean the jerk gets minimized at the cost of reaching the chord error limit), and then in Sect. 3.3 for the reward function we get a weighted combination of minimizing the chord error and violations of the jerk limit.
- How are constraints ensured when learning? In the reward function constraint violations just seem to be modeled as costs, so there is no guarantee that they don't get violated, i.e., soft constraints rather than hard constraints.
- Sect. 3.1 and 3.2: While quintic polynomial functions can indeed solve the problem for the constraints as defined by the authors, it remains very vague why that is the best idea. As far as I understood with this formulation, perfectly straight lines (which would be desirable for many mechanical parts) are not possible, and the optimized path can be quite far of. Wouldn't a higher order representation (or a different kind of spline, joining straight lines with transition 'pieces' at the corners, etc.) allow to follow the desired path more accurately? More generally: what are the implications/consequences of the design choices?
- Sect. 3.2 / Fig. 3: The result that the slower the CNC moves the more distorted paths we get is highly counterintuitive. If the objective is to avoid the limits/constraints then simply moving slower should reduce all three velocity, acceleration, and jerk. This seems to be more an artifact of keeping the boundary conditions fixed. And I also don't believe the trade-off (longer time = more traj distortion but greater velocity 'smoothness') is general - already in your plot we see that when going from T=4 to T=5 the trajectory gets more distorted, but also the velocity peak on the right (t = 3.2 and 3.7) becomes worse (i.e., the trajectory needs to accelerate drastically towards the end to achieve the boundary constraints), which seems to invalidate the claim in the paper. My feeling is that the effects will depend quite a bit on the combination of boundary conditions that the range of T you consider, and that drawing the conclusion about the trade-off based on a single example/figure isn't warranted. Simple example: Boundary conditions and T are chosen in a way that the path is a simple straight line with constant velocity (and zero acceleration and zero jerk) so zero chord error, then both increasing and decreasing T will require some non-zero acceleration and potentially require path deviations.
- Eq (11): What is the purpose of having N-i as part of the state, but no indication of which segment we are currently in?
- Eq (12): p^L and p^R are shown in Fig 2 but don't seem to be explained in the text. Nor does it become clear how far away they can be from p. \delta_max? But then how do you ensure that the curved segment in between doesn't protrude outside that limit?
- l. 295: I assume the absolute value of the jerk j_i(t) should be used in r_i^jerk
- Algo 1: only shows policy execution, I think it would be interesting to also show the training procedure
- Sect. 4.1: "due to unavailability of baseline implementations" for PPO and SAC there are quite a few implementations available (e.g. Stable Baselines), or do you mean something else?
- Sect. 4.2: Why is the chord error not evaluated/reported? I think this would be crucial to see at which accuracy cost the improved other metrics come. E.g. in the Fig 5 inset, the path error seems to have increased quite a lot - if we want to have accurate points.
- I'm not from the CNC field, but are these decorative shapes really representative for many tasks? I'd assume for more technical applications accurate straight lines and sharp corners are actually crucial.
- Table 1: It is unclear what we see here. Single RL run? For RL papers it is crucial to report statistics over several runs (mean + std).
- Sect. 4.3: I really would have liked to see some ablations on your method (rather than only 2 different RL algorithms) and sensitivity analysis on the various parameters there are in the approach (e.g. reward weights).
- Sect. 4.4: The state and action space was designed with generalization in mind, now figuring out that it needs to be retrained for all paths after all is disappointing. Related to this, it would have been nice to see some results on the performance without retraining.
- A few typos, e.g. l. 140 "udden"

---

### Official Review · Reviewer_Tp6q · 2024-11-02

**Soundness:** 3
**Presentation:** 3
**Contribution:** 2
**Rating:** 3
**Confidence:** 3

**Summary:**

This paper proposes reinforcement-learning based method to solve the integrated integrated toolpath smoothing
and feedrate planning problem in CNC machining. PPO and SAC are used to train RL agents to predict intermediate kinematic states. To generate the trajectory, the target is to perform an integrated optimization to find a trajectory that minimizes a weighted sum of both the trajectory jerk (related to smoothing of the path) and the machining time, taking into account kinematic constraints. Then RL finds intermediate kinematic states at the path segment boundaries. The method is evaluated on four tool paths and shows generally better performance than the benchmark methods used in the evaluation.

**Strengths:**

Applying RL-based methods for improvement of trajectory tracking in CNC machining is novel for this applicaiton domain. The approach is original and clearly explained in the paper. I appreciate the provided algorithm and the paper's presentation in general.

**Weaknesses:**

1) The authors solve the simultaneous problem of smoothing a path given by linear segments and optimizing the feed rate (velocity) of the CNC machine. In their literature study, they omit existing work on this problem (see references). The problem can be seen in two different ways: 1) improving the performance of the system, by improving both the time duration and the position error; and 2) solving mathematically the smoothing problem together with feed rate optimization, which is the preferred one in this paper. For both problems, there is relevant literature which is omitted. I provide a couple of references, addressing both.  It would be useful to include that in the literature review.

Zhang, Y., Wang, T., Peng, P., Dong, J., Cao, L. and Tian, C., 2021. Feedrate blending method for five-axis linear tool path under geometric and kinematic constraints. International Journal of Mechanical Sciences, 195, p.106262.

Liu, B., Xu, M., Fang, J. and Shi, Y., 2020. A feedrate optimization method for CNC machining based on chord error revaluation and contour error reduction. The International Journal of Advanced Manufacturing Technology, 111, pp.3437-3452.

Kim, H. and Okwudire, C.E., 2020. Simultaneous servo error pre-compensation and feedrate optimization with tolerance constraints using linear programming. The International Journal of Advanced Manufacturing Technology, 109, pp.809-821.

A. Rupenyan, M. Khosravi and J. Lygeros, "Performance-based Trajectory Optimization for Path Following Control Using Bayesian Optimization," 2021 60th IEEE Conference on Decision and Control (CDC), Austin, TX, USA, 2021, pp. 2116-2121, doi: 10.1109/CDC45484.2021.9683482.


2)  Even when the preferred way of addressing the performance problem is somoothing+feed rate optimization, some quantification of the tracking error is missing. It would be useful to see what is the effect of the approach on the positioning performance.

3) While there is a comparison with some approaches treating the feed rate and the path generaiton separately, there is no comparison with approaches treating the problems jointly (see references above). If such a comparison is added, I would be willing to increase my rating of the paper.

4) There is no quantification or discussion of the computational performance of the method. Is it intended to be used offline?

**Questions:**

1. In the simulations, somethimes the KIRL-PPO shows better results, and sometimes the KIRL-SAC. Are there any criteria to decide when to use PPO or SAC?

2. Does the method guarantee that there is no constraint violation?

---

### Official Review · Reviewer_hemx · 2024-11-03

**Soundness:** 2
**Presentation:** 3
**Contribution:** 2
**Rating:** 3
**Confidence:** 4

**Summary:**

This paper proposes a reinforcement learning (RL) method called KIRL for integrated joint toolpath smoothing and feedrate planning in Computer Numerical Control (CNC) machining. The tool trajectories are divided into segments by a series of boundary points and quintic polynomial functions between them. The RL agent is trained for predicting kinematic states at the boundary points, which are then used for polynomial interpolation. The duration of each segment is separately optimized by maximizing the reward function. Experimental results demonstrate that KIRL can generate smoother trajectories and optimize machining time compared to traditional decoupled methods.

**Strengths:**

The problem this paper aims to solve, i.e., integrated toolpath smoothing and federate planning, is rooted in real-world manufacturing. It is important for improving machining accuracy, efficiency, and tool life. The proposed method of using RL for trajectory optimization is original, and its performance is better than traditional decoupled approaches. The writing of the paper is clear.

**Weaknesses:**

1. The paper does not convince me why the kinematic state prediction problem is an MDP. In particular, why does it have the Markov property? The authors defined an observation space, but what they need to define is a state space that makes the problem an MDP. From my understanding, some elements in the current observation space already violates the Markov property. For example, the segment length and turning angle in the next step cannot be determined by observation or action at the current step. The authors are suggested to explicitly justify how their formulation satisfies the Markov property, and to clarify the distinction between their observation space and the underlying state space of the MDP.

2. The duration of each segment is computed by minimizing the reward function of that segment. This is not optimal because the objective is to minimize the total machining time, which should be a joint minimization on the sum of durations of all segments. From my understanding, the authors do separate optimization because solving future time duration requires future kinematic states, which are not available at the current step. If this is true, it reinforces the doubt whether the problem is an MDP because now the reward function also depends on future states. In addition, the authors are suggested to discuss the trade-offs of their approach versus joint optimization, and to clarify how their method approximates or relates to the global optimum.

3. The authors mentioned that there are some recent studies formulating the integration of toolpath smoothing and federate planning as a holistic problem, but they did not explain how these methods solve the problem, nor did they compare the proposed method with them in the experiments. It is unclear whether and why the proposed method is superior to existing integrated methods. The authors are suggested to include a brief overview of how existing integrated methods work, and a comparative evaluation against at least one state-of-the-art integrated approach.

**Questions:**

1. Why is the kinematic state prediction problem an MDP? Why does it have the Markov property? What is the state space of this MDP?
2. Why is the duration of each segment optimized separately? Is it possible to jointly optimize the sum of durations of all segments? Is the reward function still well-defined in this case?
3. How do existing integrated methods solve toolpath smoothing and federate planning? Why is RL superior to these methods? How does the proposed method perform compared to these integrated methods?

---

### Official Review · Reviewer_uu6z · 2024-11-04

**Soundness:** 2
**Presentation:** 2
**Contribution:** 2
**Rating:** 5
**Confidence:** 4

**Summary:**

This paper proposes a RL approach to improve smoothness and efficiency in following CNC machining toolpath. In CNC machining, the toolpath is defied as a G01 code which is a series of points. Path between intermediate points are straight line segments connecting them. The junctions introduce discontinuity in velocity and acceleration. Traditional approaches first smooth the trajectory, then adjust the tool path velocity to accommodate maximum velocity, acceleration and jerk constraints. This de-coupling can introduce inefficiencies. Hence, the authors propose a coupled optimization approach leveraging RL.

**Strengths:**

Interesting idea of using RL to solve CNC routing problem which is traditionally solved by hand crafted algorithms

**Weaknesses:**

1. Interesting study but not enough evidence to show usefulness. Given the topic of the paper, it either demands real world results (e.g. higher quality CNC output, faster toolpath, etc) or very strong evidence in simulated results.
2. No real world results. Simulated results are also not very strong. E.g. authors mention in Line 43-45"..decoupled approach often yields suboptimal results..limiting the achievable feedrate". However, in 2/4 toolpaths, the proposed method generates slower toolpath than existing methods.
3. Given the additional time and complexity with RL based optimization, one would use it only if there is a strong reason, which is missing in the current paper.

Minor:
1. 140-141: udden->sudden
2. 157-158: I believe there is a missing bracket

**Questions:**

The time component of the optimization is removed when we move from Equation 5 to Equation 7, is it intentional and why?

---

### Note · Authors · 2024-11-12

I have read and agree with the venue's withdrawal policy on behalf of myself and my co-authors.